# Determinants of Sustainable Profitability of the Serbian Insurance Industry: Panel Data Investigation

**Željko Vojinović [1], Sunčica Milutinović [1,*] , Dario Sertić [2] and Bojan Leković [1]**

1   The Faculty of Economics in Subotica, University of Novi Sad, 24000 Subotica, Serbia;
    zeljko.vojinovic@ef.uns.ac.rs (Ž.V.); bojan.lekovic@ef.uns.ac.rs (B.L.)
2   Milenijum Insurance, 11070 Belgrade, Serbia; dario.sertic@milenijum-osiguranje.rs
*   Correspondence: suncica.milutinovic@ef.uns.ac.rs; Tel.: +381-24-628-146

**Abstract:** This paper aims to investigate the main drivers of sustainable profitability trends in the Serbian insurance industry over the years 2008–2019 (inclusive). Our study is motivated by the fact that insurance companies contribute to economic growth, and thus it is essential to understand the factors that contribute to their financial strength and stability. We use a set of standard panel regression models, including the mixed-effects model, followed by a more robust GMM estimation to uncover the linkage between selected micro-specific, macroeconomic, and institutional factors, and return of assets (ROA) and return on total premiums (ROTP). The present paper constitutes a significant contribution to the existing literature on the account of its comprehensiveness both in terms of the institutional datasets that we use, and in terms of the methodologies we apply (in particular, mixed effects and the generalized method of moments (GMM)). The estimated parameters are model-specific, and we find that firm size, GDP, the population growth rates, political stability, and the degree of specialization (in some empirical models) all lead to higher profitability. On the other hand, we observe that excessive risk-taking and inflation (in some specifications) are inversely related to profitability. Finally, we note that regulatory quality, average wage, and life expectancies are found to be not statistically significant. Accordingly, we argue that a profitability-centric managerial strategy should be based on expanded market share and stringent risk management protocols. At the macro level, we conclude that pro-growth and pro-population policies, combined with a well-oiled institutional setting that ensures political stability, constitute the best possible prescription for strong operational performance and profit sustainability in the Serbian insurance industry.

**Keywords:** insurance industry; profitability determinants; panel data estimation; GMM

## 1. Introduction

Risk is a natural satellite of everyday reality, whether we talk about business or private life. Although insurance professionals are highly competent in understanding the entire world of risks, both individuals and households (in some cases even governments) that use insurance services are important stakeholders in this game. Apart from customer-related interests, the stability of the insurance sector is among the top priorities of the national regulatory bodies. Since the insurance sector plays an important role in financial intermediation, national regulators predetermine the performance measures (especially liquidity standards and capital requirements), and usually create an early alarm system to prevent potential failures. Namely, insurance market instability (liquidity crises, massive losses, etc.) can trigger financial sector disturbances and negative spillover effects; thus, it is obvious that the insurance industry requires special supervisory treatment [1].

At the company level, the middle and top-level managers are also interested in exploring the main sources of profitability, both at the micro and macro level. At the micro level, their interests span to those business segments that create expenditures and revenues, aiming at optimizing the risk profile and consequent cash flow. At the macro level, they are

interested in understanding the extent to which the leading macroeconomic trends affect the financial conditions of their companies. The main goal here is to establish a dynamically optimal investment strategy allowing permanent portfolio revisions, rebalancing, and reinvesting, with the ultimate objective of taking advantage of ongoing investment opportunities or mitigating negative financial shocks.

Another way of understanding the topic is to understand a well-known conflict of interest between different stakeholders: insurers, insured agents, regulators, and the government. This makes the profitability analysis even more complicated and challenging, which is best articulated in the famous statement: "Measurement of profitability is to some extent, like beauty, in the eye of the beholder." [2]. This reveals the fact that any meaningful profitability analysis must consider many dimensions of the insurance business. To investors and insurers, profitability has a golden ring to it. To the policyholders of a stock insurer, it sounds like markup, while to those insured by a mutual company it is neutral. Insurance regulators either encourage profitability, when concerned with solvency, or seek to curtail it, when regulating rates. The IRS seeks to inflate it and consumer groups seek to minimize it. Having in mind such a challenging task, we have formulated a couple of research hypotheses that will be tested in the empirical part of the paper:

**Hypothesis 1 (H1).** *The firm-size matters for profitability.*

**Hypothesis 2 (H2).** *The insurance companies exploit the market power to boost their profitability through the economies of scale.*

**Hypothesis 3 (H3).** *GDP and political stability contribute to the profit rates of the insurance companies in Serbia.*

The main goal of the study is to investigate the effects of the selected macroeconomic and firm-specific factors on the profitability dynamics, and to create an appropriate strategy to ensure sustainable profitability in the long run.

The profitability analysis in former transitional countries such as Serbia is of particular importance, having in mind the rapid structural changes the insurance sectors lived through during the ownership transformation (2000–2007). Many state-owned insurance companies were liquidated, a new regulatory framework was established, and the insurance market became highly internationalized (more than 60% of the market is controlled by foreign insurers). The market went through the global financial crisis (2007–2008) successfully due to strict risk-assessment standards and robust investment strategies with quick hedging across different financial markets. In the observed period (2008–2019), there were 18 insurance companies (100% of the market), and there were many interesting development trends to be emphasized: (a) rapid internationalization (75% of the market is controlled by foreign companies); (b) the rise of non-life insurance (particularly in the agricultural sector); (c) the rise of voluntary health insurance; (d) persistently high market concentration (70–80% of the market is controlled by the top five companies); (e) major investments in public debt instruments (92% of total investments). Out of 18 companies included in the sample, 4 companies are exclusively focused on life insurance and 6 companies are dealing with non-life insurance, whereas the other companies are focused on both types of insurance (including reinsurance). The insurance sector in Serbia is currently quite stable (77.6% non-life insurance premium vs. 22.4% of life insurance premium), with the growth rate of the total asset between 5% and 7% on average, and a very high liquidity ratio (liquid assets/liquid liabilities is above 150%).

The literature dealing with the profitability determinants of the Serbian insurance market is quite scarce, and the analysis is mainly focused on descriptive analysis and ordinary estimation approaches. In contrast, the study offers a more sophisticated mixed-effects estimation (assuming that the between-effects and within-effects could be different), followed by a dynamic GMM panel data analysis. The results of the empirical analysis are model-specific, but it is possible to outline the most important general empirical findings.

Namely, Serbian insurance companies exploit the economies of scale, whereas excessive risk-taking practice (and inflation in some specifications) is negatively associated with profitability. We also find that economic growth and population growth (serving as a proxy for insurance market growth) are positively associated with profitability. Finally, political stability is positively associated with the profitability of the Serbian insurance industry.

Structure-wise, the paper introduces the research topic and lists the most important empirical studies in the introductory part and literature review, respectively. It is followed by a brief outline of the research methodologies implemented in the present study, including the sample structure, sources of data, and expected results. The next part discusses the most important empirical results and compares them with the most relevant studies in the field. The final section of the paper concludes, providing the potential limitations of the study, as well as some directions for future research.

## 2. Literature Overview

Many recent empirical studies are devoted to analyzing the profitability determinants of insurance companies worldwide. Methodology-wise, most of the studies dealt with both static and dynamic panels mainly covering the post-crisis period (2008 and onward). In addition, researchers mainly implemented empirical investigations, with multiple model specifications and both country-based and regional-based data samples. Therefore, it is not surprising that the results are quite different and even conflicting in some cases.

The first impression regarding implemented methodologies is related to the nature of the profitability determinants. Namely, some researchers paid particular attention to the macro-specific factors, although most of them deal with the micro-related profitability drivers. To start with, it is worthwhile to mention that some authors found interest rate, income, unemployment, and stock market movement to be important profitability determinants in the US insurance sector [3]. In addition, a mixed-factor-based investigation emphasized that unexpected inflation and interest rate, apart from liquidity and underwriting profit, are the main profitability drivers in the UK [4]. More recently, Ortyński has revealed that that the profitability of insurers in Poland is positively associated with firm size and GDP, while there is a negative relationship between underwriting risk and operating expenses [5].

Many other empirical studies consider other internal forces, such as portfolio diversification and operational performances important for financial strength of insurance companies. For example, a very extensive study by Browne, Carson, and Hoyt [6] highlights adverse financial effects of portfolio diversification on the US insurance market [6]. On the other hand, the insurance market rewards a significant discount to diversified insurers, offsetting these negative financial repercussions. However, comparable studies that focus on the EU insurance sector are rather controversial on this topic. Namely, some authors did find a positive and significant relationship between portfolio restructuring and profitability; moreover, these effects are accelerated by the cost-cutting strategy and the premium growth strategy [7]. In addition, the insurance sector largely benefits from internationalization of the non-life insurance market segment and accelerated economic growth as a consequence of FDI net inflows.

There are some very interesting empirically oriented studies that tackle insurance markets in developing countries of Asia and Eastern Europe. Namely, Chen and Wong [8] had investigated the selected insurance industries in Asia, and found that portfolio performances, size, asset, and product mix play an important role in generating profit. At a country level, Almajali, Sameer, and Al-Soub [2] investigated the publicly listed insurance companies in Jordan, revealing a positive and significant relationship between firm size, management skills, leverage, and profitability. In three independent studies, Bawa and Chattha [9] and Mulchandani, Sitlani, and Mulchandani [10] investigated the profitability determinants of life insurers in India. Overall, with an increase in capital, leverage, and commission fees, one would expect profitability to be decreasing. At the same time, firm size, liquidity, solvency, and underwriting risk are positively associated with profitability.

In addition, Charumathi [11] scrutinizes the Indian insurance market regarding what firm-specific factors might contribute to greater profitability. The author emphasizes that firm size and liquidity are positively associated with profitability, whereas leverage, premium growth, and equity are negatively associated with profitability. In the end, Jerene [12] finds a positive relationship between capital adequacy, GDP, and profitability, and a negative relationship between liquidity, inflation, and profitability.

There are a couple of profitability studies related to some other Asian countries, including the insurance industry in Pakistan (see [1,13,14]). According to these studies, profitability is negatively related to loss ratio, leverage, and earnings volatility. On the other hand, it is positively related to firm size, risk, age, and capital, whereas liquidity is not statistically significant. In addition, Dogan [15] and Öner Kaya [16] investigate profitability trends of the Turkish insurance companies. The first study reveals a positive relationship between ROE and size, and a negative relationship between ROE, age, liquidity, loss, and leverage. The second paper finds that profitability is negatively associated with age and loss ratio, and that liquidity has negative effects. Finally, Lee [17] finds that leverage is negatively related to ROA, while market share is negatively related to the operating ratio of the insurance companies in Taiwan. In addition, significant profitability determinants are underwriting risk, reinsurance, input costs, and holding-group membership.

More importantly, there are some European empirical studies that are worth noting, especially the ones dealing with Eastern European countries. For example, Hrechaniuk et al. [18] examine the firm-specific profitability factors of the insurance sectors in Spain, Lithuania, and Ukraine. They revealed a positive association between the premium growth rate and profitability, although the relevance of the study is questionable to some extent. Namely, the Spanish and Lithuanian companies operate on the single EU market. On the other hand, the insurance companies in Ukraine serve a significantly different market segment, including a much different economic and political setting they are faced with. In addition, Pervan and Pavic-Kramaric [19] focus on the non-life insurance companies in Croatia, investigating both internal and external profitability determinants. They confirm that ownership structure, operative efficiency, and inflation are positively related to profitability, whereas past profitability significantly contributes to current profitability. In addition, Curak, Pepur, and Poposki [20] investigate the main profitability drivers of the composite insurers in Croatia. They conclude that firm size, risk-exposure, and inflation are among the most important profitability determinants, though the market itself is dynamic and underdeveloped. Finally, Curak, Utrobicic, and Kovac [21] suggest that leverage, size, ROI, ownership, and the non-life insurance market share are the most important determinants of reinsurance profitability in Croatia.

We also find very interesting empirical studies dealing with the determinants of the financial performances in the European Union. For example, Valaskova, Kliestik, and Garjdosikova [22] implement a nonparametric one-way analysis of variance to explore whether the set of indebtedness ratios for a sample of 779 Slovak and Czech companies is the same across countries, districts, and firm sizes. It seems that the total indebtedness ratios and the self-financial ratios differ significantly across regions, size-groups, and legal forms of the companies. In a similar vein, Belas et al. [23] investigate how entrepreneurs from production and transportation sectors in the Slovak Republic and the Czech Republic perceive their business environment based on the industry in which their companies operate. The entrepreneurs consider the business environment favorable for starting a business, though it is reasonably risky (there is no correspondence in the case of Slovak Republic). There is also general agreement with the statement that the business environment has improved significantly over the past five years (with a neutral position in the case of the Slovak production sector). In addition, Kliestik et al. [24] predict the future financial development and dependence between selected financial ratios and the origin of a particular enterprise within CEE countries. The study finds significant differences in preferred ratios and methodologies among these countries, including different consequences of common political, market, and economic conditions within each group of nations [24]. Finally, Fidanoski et al. [25] investigate the profitability determinants of the banking industry in

Croatia, using a dynamic panel data econometric framework. The authors find that bank size, loan portfolio, leverage, and GDP are positively related to the profitability, whereas risk-exposure and administrative costs are negatively related to the profitability. Basically, Croatian banks should improve operative efficiency and risk management practice to increase their profitability [25]. Finally, the regulatory authority in Croatia should impose some additional antitrust measures to increase competition in the banking market [25].

The three studies that are more closely related to this research deal with the insurance sectors in Bosnia and Herzegovina, Romania, and Serbia. Specifically, Pervan, Curak, and Marjanovic [26] analyze the main profitability determinants of the insurance industry in Bosnia and Herzegovina. The results argue against the home-field advantage hypothesis, because the domestic companies are outperformed by the foreign ones. In addition, market share, age, and past business performances are the key driving forces of profitability, while portfolio diversification is not statistically significant. In addition, Burca and Batrinca [7] specified both fixed-effect and random-effect profitability models, aiming at investigating the micro-specific profitability factors in Romania. It seems that solvency margin, retention ratio, and firm size are positively related to profitability, whereas leverage and underwriting risk are negatively related to profitability. Finally, Kocovic, Paunovic, and Jovovic [27] investigate the micro-specific profitability factors among the non-life insurance companies in Serbia in the period 2006–2012. Their findings suggest that firm size, investment ratio, and premium growth are positively associated with profitability; also, leverage, retention rate, and combined ratio are negatively associated with profitability.

Lastly, we have been enduring the pandemic since March 2019, and there are some novel studies dealing with financial conditions of the insurance companies in different regions. For example, Babuna et al. [28] deal with the impact of COVID-19 on insurance using survey data of 55 life and non-life insurance companies in Ghana. They conclude that the crisis had detrimental effects on the insurance sector, followed by a significant drop in profitability ($-16.6\%$) and total premium (17.01%). In addition, Shevchuk et al. [29] explore how this new economic reality caused by COVID-19 is related to financial performances of the insurance companies in Ukraine. It seems that the new market landscape is based on a new CRM, which could be the catalyst for innovation in the insurance business. In addition, Goodell [30] outlines that the financial markets, including the global insurance industry, suffer from an exogenous shock caused by the pandemic-induced events. Another event-based study, by Farooq et al. [31], is slightly more market-oriented since it deals with the impact of COVID-19 on stock market returns of selected insurance firms operating in developing countries. It is found that firm size, systematic risk, price-earnings ratio, profitability, and dividend yield all affect the intensity of abnormal returns in response to COVID-19 but in different event windows [31].

Furthermore, Pulawska [31] investigates profitability of the European insurance sector in light of recent COVID-19-driven economic and financial shocks. It seems that the return on assets decreased in the case of German and Italian insurance companies, whereas the solvency ratio decreased in the Belgian, French, and German insurance sectors [32]. In addition, Nehrebecka [33] develop a stress-testing framework to investigate the impact of COVID-19 scenarios on non-financial firms' probability of default using a combination of a microeconomic and a macroeconomic model. In the case of a negative scenario, the balance sheet of all banks will deteriorate significantly, mainly due to a decrease in the quality of loan portfolio. The shock spreads around to all industries, but the author concludes that the service industry is the most vulnerable [33]. Finally, Nehrebecka [34] investigates the impact of climate change risk on the quality of loan portfolio of the firms in Poland, using the sectoral module and the company module tools. The author finds that the average direct carbon tax (EUR 75/t $CO_2$) is associated with increased expenditures and reduced sales, thereby causing the profit rate to decline. In addition, the probability of default increases to a range between 6.31% and 10.12%, compared to the end of 2020 as a baseline scenario [34].

Compared to the existing literature (with not as many studies related to the Serbian insurance market), there are many obvious contributions of the present study. First, we focus on the Serbian insurance market since this particular segment of the financial market in Serbia is greatly under-researched. The study extends the current literature body by implementing a form of mixed-effects model and GMM (both first-differenced and dynamic system GMM). Finally, we consider not only the firm-specific (firm size, risk-exposure, etc.) and macroeconomic (GDP, inflation, average real wage, etc.) determinants of profitability, but also select institutional determinants, such as regulatory quality and political stability.

## 3. Material and Methods

The study covers the period from 2008 to 2019 (inclusive) using a balanced panel dataset for 18 Serbian insurance companies (a total of 216 observations). A comprehensive list of the variables, including their calculation and expected relationship with the dependent variable, is presented in Table 1. We have chosen to focus on this period because it is the period with the most rapid transition, real structural transformation, and internationalization of the Serbian insurance industry. Additionally, the period also includes the initial years of the global financial crisis, which affected the insurance companies in Serbia significantly. The micro-specific data are collected from the National Bank of Serbia (NBS) official insurance sector reports, while economywide data are mainly extracted from the international sources (The World Bank Database, UNECE). The economywide indicators include not only macroeconomic indicators such as GDP, inflation, and monthly wages, but also certain demographics (for example, population and life expectancy) and institutional variables (for example, regulatory quality and political stability).

**Table 1.** Dependent and independent variables.

| Variable | Mark | Calculation | Relationship |
|---|---|---|---|
| Return on Asset | ROA | Net-profit/Total Asset | Dependent |
| Return on Total Premium | ROTP | Net-profit/Total Premium | Dependent |
| Size | SIZE | Log (Total Asset) | + |
| Risk Exposure | RISK | Technical Reserves/Total Premium | − |
| Specialization | DUMMY | 1 for life and 0 for non-life insurance | + |
| Herfindahl-Hirschman index | HHI | Sum of squared market shares of all firms | + |
| GDP | GDP | Log of GDP per capita | + |
| Inflation | INFL | Growth rate of CPI | ± |
| Wage | WAGE | Log of gross average monthly wages | + |
| Population | POPUL | Population in Serbia | + |
| Life expectancy at birth | LEXP | Log of life expectancy at birth, total (years) | − |
| Quality of Regulation | REG | Log of percentile Rank | + |
| Political Stability | STAB | Log of percentile Rank | + |

**Source**: Authors.

The sample structure includes 18 insurance companies: 8 companies are focused on both life and non-life insurance (including reinsurance), 6 companies are dealing with non-life insurance exclusively, and only 4 companies included in the sample are exclusively focused on life insurance. These 18 companies controlled the entire insurance market in Serbia during the observed period (2008–2019). Note that there are two insurance companies that are newly established in the last couple of years, but the research sample still covers about 95% of today's Serbian insurance sector.

Methodology-wise, the study includes several panel data tools, including pooled OLS, fixed effects, and random effects. Additionally, we estimate a random intercept and trend model with time-varying covariates (within-subjects (WS) and between-subjects (BS)

effects). It is initially assumed that WS $=$ BS; then, we assume that WS $\neq$ BS. Having in mind our research agenda, we have specified two comparable profitability models as follows:

**Model 1.**

$$\text{ROA}_{i,t} = \beta_0 + \beta_1 \text{SIZE}_{i,t} + \beta_2 \text{RISK}_{i,t} + \beta_3 \Delta \text{DUMMY}_{i,t} + \beta_4 \text{HHI}_{i,t} + \beta_5 \text{GDP}_{i,t,} + \beta_6 \text{INFL}_{i,t} + \beta_7 \text{WAGE}_{i,t} + \beta_8 \text{POPUL}_{i,t} + \beta_8 \text{LEXP}_{i,t} + \beta_8 \text{REG}_{i,t} + \beta_8 \text{STAB}_{i,t} + e_{i,t,} \tag{1}$$

**Model 2.**

$$\textbf{ROTP}_{\textbf{i,t}} = \boldsymbol{\beta_0} + \boldsymbol{\beta_1}\textbf{LIQUID}_{\textbf{i,t}} + \boldsymbol{\beta_2}\textbf{LOSS}_{\textbf{i,t}} + \boldsymbol{\beta_3}\boldsymbol{\Delta}\textbf{MPR}_{\textbf{i,t}} + \boldsymbol{\beta_4}\textbf{PRODUCT}_{\textbf{i,t}} + \boldsymbol{\beta_5}\textbf{RISK}_{\textbf{i,t}} + \boldsymbol{\beta_6}\textbf{SIZE}_{\textbf{i,t}} + \boldsymbol{\beta_7}\textbf{DUMMY1}_{\textbf{i,t}} + \boldsymbol{\beta_8}\textbf{DUMMY2}_{\textbf{i,t}} + \textbf{e}_{\textbf{i,t,}} \tag{2}$$

where $e_{it}$ represents the error-term composed of the firm-specific ($\eta_i$) and time-specific effects ($\lambda_i$), followed by the time-varying error term ($\epsilon_i$). Model diagnostics include the Breusch–Pagan LM test and the Pesaran CD test of cross-sectional independence (H0: residuals across entities are not correlated), the Breusch–Godfrey/Wooldridge test of serial correlation (H0: no serial correlation), and the Breusch–Pagan test of heteroskedasticity (H0: homoskedasticity).

To address the potential endogeneity problem, we proceed to estimating GMM (both the first-differenced and dynamic system GMM). The corresponding matrix of instruments for the lagged difference is based on a diagonal combination of the time-ordered dependent variable. The moment conditions are defined as $\mathbb{E}[m(W`_i, \Delta \varepsilon_i)] = 0$, and the corresponding GMM estimator (the matrix form) is formulated as follows:

$$\hat{\lambda}^{\text{GMM}} = (G`ZS_N Z`G)^{-1} G`ZS_N Z`\Delta y,$$

where $\hat{\lambda}^{\text{GMM}} = \left[ \hat{\gamma}^{\text{GMM}}, \hat{\beta}^{\text{GMM}} \right]$, $G = (\Delta y_{t-1}, \Delta X)$, $Z = (W, \Delta X)$, while $S_N$ is an optimal weighting matrix that maximizes the covariance matrix of $\hat{\theta}$. Following our empirical setting, the model can be represented as follows: $\Delta \text{Profitability}_{it} = \gamma \Delta \text{Profitability}_{it-1} + \beta`\Delta X_{it} + \Delta \varepsilon_{it}$, where $\Delta X_{it}$ is the first-differenced matrix of independent variables, and $\Delta \varepsilon_{it}$ is the first-differenced error term.

The estimation procedure includes the descriptive analysis (the descriptive statistics and the correlation matrix) followed by multiple stationarity tests (the Levin, Lin, and Chu test, the Im, Pesaran, and Shin W-statistics, the ADF—Fisher chi-square, and the PP—Fisher chi-square). We calculated the variance inflation factor (VIF), partial R-squared, and tolerance to address multicollinearity. After estimating POLS, fixed-effects, and random-effect models, we implemented the LR and the Housman tests to detect the most consistent estimations. Finally, we estimated a model (Model 3) where $X_{it}$ (the design matrix) assumes the BS and WS effects to be equal ($\beta_{BS} = \beta_{WS}$), followed by a model (Model 4) where $X_{it}$ (the design matrix) assumes unequal BS and WS effects ($\beta_{BS} \neq \mu \beta_{WS}$). Finally, we estimated a dynamic GMM, which is additionally challenged by the Sargan test of overidentifying restrictions and the Arellano–Bond test for zero autocorrelation in differenced errors.

## 4. Results and Discussion

To open the results section, we will first provide some introductory thoughts regarding the Serbian insurance market development trends. Specifically, the market has undergone transitional changes since 2000, and the whole process was characterized by massive privatization and internationalization. In the first stage, the state-owned companies were sold, and more efficient international strategic investors entered the game. The insurance market is not remarkably developed, having in mind that the total premium to GDP ratio is only 1.9% (comparatively speaking, in the EU it is 8.25%). In addition, the market is reshuffling continuously, but it still dominated by a couple of giant players that control the market implicitly (CR5 is around 70%). Profitability-wise, the indicators have been highly influenced by the recent crisis, and the profitability trends are above the EU standards

because of suboptimal competition. Specifically, the average ROA and ROTP for the period 2008–2016 were 2.17% and 3.43%, respectively.

The descriptive analysis (see Appendix A, Table A1) shows that HHI, WAGE, STAB, and the population vary the most, whereas ROTP is significantly higher than ROA. In addition, the correlation matrix (see Appendix A, Table A2) reveals only a moderate level of interdependence between the independent variables; thus, it is highly unlikely that the model suffers from multicollinearity. To check this further, we present the results of the collinearity tests—the variance inflation factor (VIF) and tolerance in Table 2. As a rule of thumb, if the VIF exceeds 10, the model might suffer from multicollinearity due to a strong co-movement between the variables. Table 2 on the previous page shows the partial R-squared, variance inflation factor and tolerance. Using the common standards for partial R-squared (0.90) and VIF (10), it seems the model does not suffer from collinearity. A closer look at the dynamics of the profitability indicators (see Appendix A, Figure A1) shows that ROA, ROE, and the profit rate change in the same direction, with a downward sloping trend from 2008 (because of the global financial crisis), and an upward sloping trend starting from 2013. The insurance companies with higher ROA tend to have higher ROE, meaning that leverage does matter for profitability. The reinsurance market segment was more profitable before 2013, and less profitable afterwards; on the other hand, life insurance companies had much higher profitability indicators comparing to non-life insurance companies (excluding the period 2012–2013).

**Table 2.** Multicollinearity testing.

| Variable | $R_j^2$ | $VIF_j$ | Tolerance | Description |
|----------|---------|---------|-----------|-------------|
| SIZE | 0.1914 | 1.2367 | 0.8086 | No multicollinearity |
| RISK | 0.5395 | 2.1718 | 0.4605 | No multicollinearity |
| DUMMY | 0.3477 | 1.5331 | 0.6523 | No multicollinearity |
| HHI | 0.5389 | 2.1688 | 0.4611 | No multicollinearity |
| GDP | 0.1968 | 1.2450 | 0.8032 | No multicollinearity |
| INFL | 0.2690 | 1.3681 | 0.7310 | No multicollinearity |
| WAGE | 0.5475 | 2.2102 | 0.4525 | No multicollinearity |
| POPUL | 0.1566 | 1.1857 | 0.8434 | No multicollinearity |
| LEXP | 0.2657 | 1.3619 | 0.7343 | No multicollinearity |
| REGULATION | 0.5267 | 2.1127 | 0.4733 | No multicollinearity |
| STABILITY | 0.6418 | 2.7919 | 0.3582 | No multicollinearity |

**Source**: Authors.

The next stage incorporates stationarity testing at both level and first difference, and the results are provided in Appendix A (Table A3). The results show that the profitability indicators ROA and ROTP are stationary at level, while all the independent variables are non-stationary at level. On the other hand, all the independent variables are stationary at first difference, meaning that we can use the original profitability indicators and first-differenced independent variables in our model specifications. The model estimation results for Model 1 are presented in Table 3.

Table 3 shows a positive and significant relationship between the size and ROA, implying that Serbian insurance companies exploit the economies of scale. This finding corresponds to the results of related empirical studies. To be more specific, Chen and Wong [8] pointed out that firm size is a highly significant factor of the general financial health of general (property liability) and life insurance companies in the U.S. and the selected developed countries. It is also supported by Curak, Pepur, and Poposki [20] and Curak, Utrobicic, and Kovac [21], meaning that firm size, as a proxy for market power, is significantly associated with the ROA of Croatian composite insurers; also, firm size is a decisive factor in the decision-making process of Croatian insurance companies to buy reinsurance. The latter findings are also in line with the significant size-effects of market premium and market share on profitability and the efficiency of Polish non-life insurers during the period of financial integration (see [35]) and the Polish general insurance

companies (see [5]). In a similar vein, the size of the insurance-holding companies in Taiwan drives their financial performance (see [17]), whereas Öner Kaya [16] confirms that Turkish non-life insurance companies also exploit the economies of scale. Other related empirical studies also suggest a significant relationship between the size and profitability of insurers in Jordan (see [1,2]), Pakistan and India (see [11,14], respectively), and Romania (see [8]).

**Table 3.** Panel estimation results (ROA).

| Variable | POLS | | Fixed Effects | | Random Effects | |
|---|---|---|---|---|---|---|
| | Coeffic | Prob. | Coeffic | Prob. | Coefficient | Prob. |
| SIZE | 1.185 | 0.015 ** | 1.119 | 0.016 ** | 0.795 | 0.000 *** |
| RISK | −0.260 | 0.048 ** | −0.288 | 0.035 ** | −0.196 | 0.002 *** |
| DUMMY | 3.217 | 0.001 *** | 2.602 | 0.000 *** | 1.956 | 0.005 *** |
| HHI | 0.320 | 0.333 | 0.255 | 0.329 | 0.224 | 0.344 |
| GDP | 0.990 | 0.037 ** | 0.783 | 0.011 ** | 0.856 | 0.000 *** |
| INFL | −0.031 | 0.113 | −0.012 | 0.101 | −0.048 | 0.216 |
| WAGE | 0.311 | 0.095 * | 0.246 | 0.145 | 0.611 | 0.256 |
| POPUL | 0.024 | 0.002 *** | 0.016 | 0.004 *** | 0.018 | 0.000 *** |
| LEXP | −1.167 | 0.241 | −1.372 | 0.141 | −1.644 | 0.429 |
| REG | −0.311 | 0.113 | −0.443 | 0.221 | −0.606 | 0.519 |
| STAB | 0.393 | 0.059 * | 0.190 | 0.040 ** | 0.218 | 0.037 ** |
| R-sq | 0.287 | | 0.214 | | 0.332 | |
| Adj. R-sq | 0.261 | | 0.186 | | 0.308 | |
| S.E. Reg. | 1.395 | | 1.464 | | 1.350 | |
| B-P LM test * | 18.4422 ($p = 0.1577$) | | 16.3341 ($p = 0.2216$) | | 14.3621 ($p = 0.2764$) | |
| Pesaran CD * | 1.4672 ($p = 0.4265$) | | 1.6431 ($p = 0.4503$) | | 1.9441 ($p = 0.4744$) | |
| B-G/W test * | 15.3562 ($p = 0.1245$) | | 11.3562 ($p = 0.2491$) | | 12.4791 ($p = 0.1922$) | |
| B-P heter * | 4.1742 ($p = 0.2361$) | | 3.2944 ($p = 0.2865$) | | 4.8573 ($p = 0.1855$) | |

* Breusch–Pagan LM and Pesaran CD test of independence, Breusch–Godfrey/Wooldridge test for serial correlation, and the Breusch–Pagan test for heteroskedasticity. Source: Authors. Note: Significance at the 10%, 5%, and 1% level are denoted by ***, **, *, respectively.

In addition, we find that uncontrolled risk exposure would lead to adverse profitability outcomes, which is supported by many other empirical studies. For example, risk-exposure is significantly negatively related to the profitability indicators of Indian and Croatian insurance companies (see [10,21], respectively), due to the market risk-premium. On the other hand, it contradicts the findings of Kokobe and Gemechu [36], who found no correlation between risk management practice and the profitability performance of insurance companies in Ethiopia due to a poor risk management strategy. Business specialization (an exclusive focus on the life or non-life insurance market) would contribute to the better financial performance of Serbian insurers, and this reaffirms and supports the strategic focus hypothesis. Comparable results by Liebenberg and Sommer [37] reveal that undiversified insurers constantly outperform diversified insurers in the U.S. Size-wise, there is a significant difference in the financial performance of American insurance companies (1% of ROA and 2% of ROE) as a result of a diversification penalty (and diversification discount). This is further supported by the fact that product diversification, even in the case of one-type insurance businesses (non-life insurance companies in Poland), would lead to undesirable profitability outcomes (see [35]).

We also found no statistical significance of inflation, which corresponds to the findings of Lee [17] that inflation does not matter for the property liability insurance industry in Taiwan. However, it contradicts the fact that inflation matters for the financial performance of the Croatian insurance industry (see [21]). It is quite surprising that the HHI also proved to be statistically insignificant in this specification, though one would expect that market concentration leads to market power and consequent superior financial performance. On the other hand, the Serbian insurance market is highly concentrated by default, which might be used as an explanation for why we did not identify HHI-related statistical significance (ceteris paribus). Our results are contrasted by the fact that market concentration, measured

by the share in total market premium, contributes to the higher profitability of Polish insurers (see [35]) and U.S. health-insurance marketplace companies (see [3]). The latter study recognizes a positive relationship between market concentration and profitability, due to the collusive behavior or greater efficiency of larger health insurance firms.

Our results also reveal the fact that the profitability of Serbian insurance companies is fueled by the economic prosperity of the country measured by the GDP growth rate. Thus, balance sheet growth, as a result of an economic boom, combined with an effective market and credit risk strategy, would contribute to their superior financial performance. Of course, many other empirical papers highlight this relationship, especially in the case of Poland (see [5,17]). In addition, Chen and Wong [8] came up with a similar conclusion when it comes to the profitability of the insurance industry in some Asian countries, especially in Singapore [8]. Furthermore, Dorofti and Jakubik [38] also found that GDP (level or lagged term) matters for the profitability of both life and non-life insurance companies in Europe. Finally, an increasing population (market size) and political stability are also positively associated with ROA. Accordingly, an aggressive market expansion strategy (coupled with an appropriate risk management practice) at the micro-level and a stable political environment and pro-growth monetary and fiscal policy would contribute to an increase in profitability. The results seem to be plausible, since the estimated residuals are approximately normally distributed (see Appendix A, Figure A2, Panel A and B), and the diagnostic tests suggest no cross-sectional dependence, no serial correlation, and no heteroskedasticity. The results of Model 2 are presented in Table 4.

**Table 4.** Panel estimation results (ROTP).

| Variable | POLS | | Fixed Effects | | Random Effects | |
|---|---|---|---|---|---|---|
| | Coefficient | Prob. | Coefficient | Prob. | Coefficient | Prob. |
| SIZE | 1.668 | 0.005 *** | 1.499 | 0.015 ** | 1.584 | 0.010 *** |
| RISK | −0.236 | 0.016 ** | −0.414 | 0.034 ** | −0.515 | 0.000 *** |
| DUMMY | 1.678 | 0.018 ** | 1.455 | 0.009 *** | 1.331 | 0.001 *** |
| HHI | 0.019 | 0.464 | 0.211 | 0.511 | 0.331 | 0.499 |
| GDP | 0.269 | 0.001 *** | 0.220 | 0.022 ** | 0.335 | 0.043 ** |
| INFL | −0.080 | 0.058 * | −0.072 | 0.077 * | 0.069 | 0.055 * |
| WAGE | 1.507 | 0.134 | 1.258 | 0.211 | 1.606 | 0.111 |
| POPUL | 0.447 | 0.000 *** | 0.616 | 0.001 *** | 0.595 | 0.000 *** |
| LEXP | −0.583 | 0.561 | −0.614 | 0.540 | −0.4663 | 0.6417 |
| REG | −0.894 | 0.235 | −0.829 | 0.409 | −0.831 | 0.273 |
| STAB | 0.249 | 0.024 ** | 0.328 | 0.032 ** | 0.159 | 0.043 ** |
| R-squared | 0.321 | | 0.298 | | 0.334 | |
| Adj. R-sq | 0.302 | | 0.278 | | 0.315 | |
| S.E. Reg. | 1.876 | | 1.662 | | 1.592 | |
| B-P LM test * | 12.4422 ($p$ = 0.1254) | | 14.3341 ($p$ = 0.1843) | | 10.3621 ($p$ = 0.0891) | |
| Pesaran CD * | 2.5461 ($p$ = 0.4953) | | 2.1001 ($p$ = 0.4821) | | 2.3566 ($p$ = 0.4906) | |
| B-G/W test * | 19.3562 ($p$ = 0.3964) | | 14.2255 ($p$ = 0.3188) | | 11.4791 ($p$ = 0.1733) | |
| B-P heter * | 2.1742 ($p$ = 0.3256) | | 2.1990 ($p$ = 0.2711) | | 2.9244 ($p$ = 0.2201) | |

* Breusch–Pagan LM and Pesaran CD test of independence, Breusch–Godfrey/Wooldridge test for serial correlation, and the Breusch–Pagan test for heteroskedasticity. Source: Authors. Note: Significance at the 10%, 5%, and 1% level are denoted by ***, **, *, respectively.

Compared to Model 1, the results of Model 2 (ROTP) are slightly different. It seems that firm size and risk exposure matter for profitability, with the former being positively related to ROTP and the latter being negatively related to ROTP. As we emphasized above, these findings are in line with the results of comparable empirical studies (see [5,20,21,26,39]). In addition, to increase profitability, the companies should specialize their business profile, serving either life or non-life market segments (see [17]). The model provides a consistent estimation when it comes to macroeconomic variables, with the exception of the inflation rate. Namely, we found a negative and significant (at the 10% level) relationship between inflation and ROTP, which is also confirmed by similar empirical studies (see [18,20,39]).

In our opinion, one of the main driving forces behind this negative relationship might be the maturity mismatch that emerges from time-sensitive insurance contracts, making the insurance companies worse-off due to proportionally greater time-sensitive liabilities. Finally, population growth and political stability are positively related to ROTP, which reinforces the need for a favorable macroeconomic and institutional environment (stable economic growth, low inflation, stable political environment, etc.).

Although we have a consistent estimation among the three estimators, we guess what results we should consider the most appropriate. We run several tests (the LR test, the F test of individual effects, and the Hausman test) interchangeably to detect the best model to be interpreted (see Appendix A, Table A4). Based on both the LM and the Hausman tests, the winner is the RE model. The estimated results can be considered plausible, since the residuals plots (see Appendix A, Figure A2, Panel C and D) support the *i.i.d.* assumption. In addition, model diagnostics based on a set of independence tests suggests that we fail to reject the null hypothesis in each specific case. It implies that there is no cross-sectional dependence, no serial correlation in the panel models, and there is no heteroskedasticity. To make the results as robust as possible, we offer another set of estimates (see Table 5) that are in line with our assumption of both FE and RE across the sample.

**Table 5.** Mixed-effects model (Model 3 : $\beta_{BS} = \beta_{WS}$).

| | Panel A: ROA | | | Panel B: ROTP | |
|---|---|---|---|---|---|
| **Variable** | **Coefficient** | **Prob.** | **Variable** | **Coefficient** | **Prob.** |
| TIME | 0.016 | 0.001 *** | TIME | 0.020 | 0.002 * |
| SIZE | 0.983 | 0.000 ** | SIZE | 0.752 | 0.005 *** |
| RISK | −0.185 | 0.004 ** | RISK | −0.222 | 0.001 *** |
| DUMMY | 1.231 | 0.006 *** | DUMMY | 1.012 | 0.000 *** |
| HHI | −0.132 | 0.011 ** | HHI | −0.331 | 0.013 ** |
| GDP | 0.422 | 0.046 ** | GDP | 0.224 | 0.021 ** |
| INFL | −0.104 | 0.017 ** | INFL | −0.081 | 0.045 ** |
| WAGE | 1.471 | 0.222 | WAGE | 1.127 | 0.346 |
| POPUL | 0.396 | 0.001 *** | POPUL | 0.291 | 0.004 *** |
| LEXP | −0.339 | 0.149 | LEXP | −0.368 | 0.414 |
| REG | −0.566 | 0.292 | REG | −0.655 | 0.301 |
| STAB | 0.313 | 0.016 ** | STAB | 0.257 | 0.023 ** |
| R-squared | 0.274 | | R-squared | 0.294 | |
| Adj. R-sq | 0.251 | | Adj. R-sq | 0.277 | |
| S.E. Reg. | 1.452 | | S.E. Reg. | 1.261 | |
| B-P LM test * | 21.4422 ($p = 0.4722$) | | | 16.3677 ($p = 0.3479$) | |
| Pesaran CD * | 1.6901 ($p = 0.4916$) | | | 2.4933 ($p = 0.4936$) | |
| B-G/W test * | 9.8835 ($p = 0.1244$) | | | 12.5431 ($p = 0.1922$) | |
| B-P heter * | 2.5531 ($p = 0.1866$) | | | 1.9943 ($p = 0.2541$) | |

* Breusch–Pagan LM and Pesaran CD test of independence, Breusch–Godfrey/Wooldridge test for serial correlation, and the Breusch–Pagan test for heteroskedasticity. Source: Authors. Note: Significance at the 10%, 5%, and 1% level are denoted by ***, **, *, respectively.

The results of the mixed-effects model (BS = WS) resemble those from the random-effects model, although there are some obvious differences. First, the profitability indicators have been increasing over time, whereas competition matters for profitability—a higher HHI (more competitive market) implies lower profitability indicators. It seems that insurance companies might be able to exploit less competitive markets through further collusive behavior and market dominance. This finding is in sharp contrast with related empirical studies (see [3,27]) and our previous estimation, meaning that the relationship between competition and profitability is not robust to different specifications. Firm size also matters for profitability, which is in line with our previous estimation and many comparable research studies (see [8,21,26]); furthermore, there is a negative and significant relationship between inflation and profitability, which is also confirmed by Chen and Wong [8]. Simply speaking, it might happen that the balance sheet maturity mismatch causes profitability indicators to decrease when the inflation rate is increasing.

As we have emphasized before, the population as a proxy for market expansion (market potential), the GDP growth rate, and political stability are positively related to profitability. There is one significant difference in the estimated ME model, and it is related to business specialization. Namely, the business specialization dummy (life vs. non-life insurance) seems to be statistically significant, which implies that life insurance companies in Serbia are more profitable than non-life insurance companies (similar to [3]). Since the non-life market insurance segment has been growing rapidly over time, there is a possibility that increasing revenues are offset by financial losses caused by poor risk-management practice, which leads to lower profitability. The latter finding and corresponding explanation is in line with a statistically significant negative relation between excessive risk-taking practice and profitability. It further supports our assumption that the young and still unstructured insurance market in Serbia actually penalizes excessive risk-taking practice.

Finally, when assuming different between subjects (BS) and within subjects (WS) effects (see Appendix A, Table A6), the results generally match our previous estimation; however, there is an opposite relationship of size and profitability when it comes to the BS effects (positive) vs. WS effects (negative). Overall, Serbian insurance companies exploit the economies of scale, which drives down unit costs and boosts their profitability (see [5]). We find the same divergent marginal effects with respect to risk exposure, market competition, inflation, GDP, population, and political stability. Thus, these variables matter for profitability indicators, but their contribution differs across different variability dimensions. It also reinforces our previous statement that the estimated marginal effects are strongly model-specific, whereas our assumption that BS might differ from WS seems to be more realistic. Comparatively speaking, these findings are more or less in line with our previous results, as well as with the estimated marginal effects of related empirical studies (see [8,20,26,27,39]). As for model diagnostics, we fail to reject the null hypothesis in both cases (both ME models), meaning that we did not find cross-sectional dependence, serial correlation in panel models, or the heteroskedasticity problem.

The GMM results (see Appendix A, Table A6) support our previous conclusions up to a certain level. To be quite specific, profitability is a self-fueling process due to the fact that positive financial performances in the past most likely affected future financial performance in the short run. Insurance companies exploit the economies of scale—as companies grow, we would expect profitability to increase due to greater cost-effectiveness (see [26,39]). Risk-exposure implies lower profitability, meaning that the insurance market does not reward excessive risk-taking practice followed by poor risk-management effectiveness. We also found that economic growth, population growth, and political stability are positively related to both ROA and ROTP. These findings are closely related to the empirical results of many comparable studies (see [18,20,39]). With that being said, we would expect the Serbian insurance companies to be better-off if the whole economy was growing, assuming a prolonged period of political stability.

Compared to our previous specifications, business specialization, market concentration, inflation, wage, life expectancy, and regulatory quality seem to be statistically insignificant at the 5% significance level. Having in mind the verified differences in the estimated results, as well as the fact that some of the selected variables are statistically insignificant, this topic requires further investigation. Based on the Sargan test of overidentifying restrictions (J-statistic), we fail to reject the null hypothesis that the instruments are valid, meaning that the results seem to be plausible. The Arellano–Bond autocorrelation tests (see Appendix A, Table A6) suggest that the first-differenced errors are first-order serially correlated, which is immanent to the model by default. On the other hand, the first-differenced errors are not serially correlated at order two, implying that the GMM results are plausible.

## 5. Strategy for Sustainable Insurance in Serbia: A Proposal

Based on the UN Environment Program Financial Initiative, sustainable insurance is a strategic approach where all the activities in the insurance value chain are carried out in a

responsible and forward-looking way by identifying, assessing, managing, and monitoring risks and opportunities associated with the environmental, social, and governance issues. It also includes many operational frameworks that are logically connected, and that will help the insurers to build an effective business model (potential for growth, acceptable rate of return, etc.) on one hand, and to contribute to the environmental, social, and economic sustainability, on the other hand. Due to the increasing social, environmental, and political pressures across the globe, insurance companies are faced with an ever-increasing pressure to demonstrate their progress on sustainability. Accordingly, insurance companies have to be ready not only to react to these challenges, but also to create changes, working proactively with different regulatory agencies. Thus, the main supporting pillar of the strategy for sustainable insurers can be viewed as a combination of strategic and operational actions, with clearly defined goals (see Table 6).

**Table 6.** Five Opportunities to improve sustainable strategy for insurers.

| | |
|---|---|
| Clarify definition and strategy | What do insurers mean by "sustainability" in practical terms? How are sustainability leaders expect to achieve their mission? |
| Moving beyond the "goodnes phase" | Many people think on sustainability as voluntarism, philantropy, and good corporate citizenship. How can CSO convince business leaders to treat ESG considerations as part of the core business strategy? |
| Establish more definitive metrics | What criteria are used to judge ESG progress and success? What performance benchmarks could be used to connect ESG efforts to top-line and bottom-line ROI. |
| Bolster CSO resources | At most insurers, sustainability is a big job entrusted to a small team. If sustainability leaders had more resources, they could pursue more impactfull internal incentives to alter products, services, investments and operating models and expand their influence with policyholders and policymakers. |
| Spend more time on transforming then reporting | Most insurer CSO spend nearly of all their time gathering information for the company's annual ESG report, responding to independent ESG assesment firms and analysts and briefing key investors. The results are often as almost on exclusive focus on compliance and communication rather than transformation initiatives. |

**Source:** Modified according to Deloitte Insights [40].

As we can see, an important step in this process is to appoint a Chief Sustainable Officer (CSO) and to empower his position with enough responsibilities and financial resources to be able to create an effective sustainable strategy. His role would be to make visible actions to make the insurance company more metrics-oriented, goal-minded, and flexible to change its products and service (or offer the new ones) for this sustainability agenda to be more realistic and feasible. The CSO would be also responsible for creating a dynamic and ever-changing business model that would provide an acceptable rate of return and serve broader social goals (to improve equality by serving those areas with pronounced environmental problems; to support gender diversity in governance bodies, etc.). To be more specific, the "governance" side should be defined not just in terms of its transparency, but also by how effectively the company is enhancing diversity and inclusion in management, executive leadership, and the board [40]. This strategic initiative

would create better corporate climate and other operational-level leaders that can address environmental, social, and governance (ESG) issues.

The situation in Serbia in this regard is far from being resolved. Namely, a narrow market does not provide enough resources to make significant investments into this sustainability strategic initiative. In addition, the nationwide regulation and internal governance practice do not recognize the importance of these changes and transitions within Serbian insurance companies. Accordingly, there would be several important steps to be completed to allow Serbian insurance companies to become more sustainability-minded enterprises: (1) to make organizational and structural changes; (2) to appoint a separate governance body responsible for the sustainable insurance practice; (3) to establish a special sustainability project team that would collect the financial resources from the EU and local resources; (4) to follow comparable practice in the EU countries, and to require a strong support from the government to focus on the sustainable insurance business practices. At the operational level, it is necessary to create specific financial incentives to insure the projects for renewable energy resources, energy efficiency solutions, organic food production, etc. For example, it is possible to offer a lower premium or government-sponsored subsidies for green economy projects, energy-efficient houses, and facilities that reduce pollution. In addition, Serbian insurance companies might want to offer some attractive insurance products to the companies that appoint at least one third of women in their executive bodies. This could be a good way of supporting gender diversity in corporate governance in Serbia. To wrap up, there are important steps to perform to place the Serbian insurance sector on the track of a sustainable insurance industry. Our empirical research identifies the profitability determinants that can be exploited to increase the rate of return on the total assets and total premiums. However, it is also necessary to incorporate this sustainable practice into an effective sustainability-based business model that will recognize not only the pure economic goals (the profit rate, the market share, risk exposure, liquidity, etc.) but also common social, environmental, and diversity and inclusion objectives.

## 6. Conclusions

This paper deals with micro-specific and economywide factors that are intrinsically related to the profitability of some 18 Serbian insurance companies (comprising 100% of the market) in Serbia, over the period 2008–2019. Our strong empirical results provide some most convincing arguments about the key fundamental underliers of profitability. Namely, we find that firm size, level of specialization, GDP, population, and political stability are positively related to profitability, whereas risk-exposures, HHI (in ME specification), and inflation tend to detract from financial performance. On this score, our GMM estimates offer the following robust results: (a) profitability is persistent and self-reinforcing; (b) inefficient risk management practices might endanger financial performance; (c) economic growth, political stability, and growing market share all contribute to profitability in a most essential way.

We believe that a successful profitability strategy must be focused on expanding market share, together with operational synergies, successful risk management practice, and greater market penetration in the life insurance segment. Such would allow firms to divide and conquer the market, increase total premiums, exploit general market trends, and boost profitability. On the account of the central role that the insurance sector plays in financial intermediation, our findings have very definite implications for macroprudential policy. Namely, the government must act to ensure prolonged, stable, and sustainable economic growth, coupled with political stability and institutional efficiencies that allow Serbian insurance companies to increase their profitability and thrive on organic growth.

Although the paper is general and comprehensive, we make it a point to highlight certain basic limitations of our study. Namely, the sample size is small (only eighteen companies), and it might be useful to add data from other insurance firms in the Western Balkans to obtain results that are even more dependable. By combining observations from heterogenous insurance markets, we might be able to find appropriate instrument(s) and

use IV estimators to address potential endogeneity problems. Additionally, the work relies mainly on accounting ratios, because market-oriented measures of financial performance such as P/E ratios or PEG ratios are not presently available (most Serbian insurers are not publicly listed).

Finally, this segment of the Serbian financial system is significantly understudied, and there are many fruitful directions for future research. For example, it would be interesting to investigate the effects of liquidity, operational efficiency, and solvency, in addition to the variables that were included above. Furthermore, a more sophisticated model might include not only typical macroeconomic variables, but also certain institutional determinants such as government efficiency, corruption, economic freedom indices, etc. In terms of methodology, a possible next step could be to explore profitability determinants using techniques of Bayesian statistical inference and modern machine learning (mainly recurrent neural network models).

**Author Contributions:** Conceptualization, Ž.V. and S.M.; methodology, Ž.V. and S.M.; software, B.L. and D.S.; writing—original draft preparation, Ž.V. and S.M.; writing—review and editing, Ž.V. and S.M.; visualization, B.L. and D.S.; supervision, Ž.V. All authors have read and agreed to the published version of the manuscript.

**Funding:** This research received no external funding.

**Institutional Review Board Statement:** Not applicable.

**Informed Consent Statement:** Not applicable.

**Data Availability Statement:** Data are not publicly available, though the data may be made available upon request from the corresponding author.

**Conflicts of Interest:** The authors declare no conflict of interest.

## Appendix A

**Table A1.** Descriptive statistics.

|  | ROA | ROTP | SIZE | RISK | HHI | GDP | INFL | WAGE | POPUL | LEXP | REG | STAB |
|---|---|---|---|---|---|---|---|---|---|---|---|---|
| Mean | 2.17 | 3.43 | 12.55 | 7.46 | 1285.0 | 4.12 | 6.37 | 666.12 | 7,204,794 | 74.78 | 52.52 | 40.41 |
| Median | 2.44 | 3.99 | 14.63 | 6.00 | 1259.5 | 4.11 | 7.30 | 653.20 | 7,199,077 | 74.80 | 53.10 | 38.90 |
| Maximum | 5.18 | 7.62 | 14.94 | 18.99 | 1637.2 | 4.14 | 12.40 | 819.60 | 7,350,222 | 75.50 | 57.20 | 55.20 |
| Minimum | 0.08 | 0.15 | 8.14 | 0.19 | 1112.7 | 4.10 | 1.10 | 561.90 | 7,057,412 | 73.90 | 45.60 | 27.40 |
| Std. Dev. | 1.55 | 2.39 | 3.06 | 6.61 | 156.19 | 0.01 | 3.89 | 76.87 | 96,828.27 | 0.60 | 3.27 | 9.68 |
| Jarque–Bera | 4.92 | 7.77 | 28.41 | 13.33 | 26.81 | 6.55 | 10.58 | 6.38 | 11.33 | 16.13 | 11.62 | 13.15 |
| Probability | 0.09 | 0.02 | 0.00 | 0.00 | 0.00 | 0.04 | 0.01 | 0.04 | 0.00 | 0.00 | 0.00 | 0.00 |
| Observations | 216 | 216 | 216 | 216 | 216 | 216 | 216 | 216 | 216 | 216 | 216 | 216 |

**Source:** Authors.

**Table A2.** Correlation matrix.

|  | ROA | ROTP | SIZE | RISK | HHI | GDP | INFL | WAGE | POPUL | LEXP | REG | STAB |
|---|---|---|---|---|---|---|---|---|---|---|---|---|
| ROA | 1.00 |  |  |  |  |  |  |  |  |  |  |  |
| ROTP | 0.99 * | 1.00 |  |  |  |  |  |  |  |  |  |  |
| SIZE | 0.06 * | 0.07 * | 1.00 |  |  |  |  |  |  |  |  |  |
| RISK | −0.05 * | −0.04 * | 0.46 * | 1.00 |  |  |  |  |  |  |  |  |
| HHI | 0.01 * | −0.01 | 0.31 | 0.46 | 1.00 |  |  |  |  |  |  |  |
| GDP | 0.31 * | 0.35 * | 0.38 * | 0.39 * | 0.50 | 1.00 |  |  |  |  |  |  |
| INFL | −0.05 * | −0.05 * | 0.49 * | 0.51 * | 0.46 * | 0.43 * | 1.00 |  |  |  |  |  |
| WAGE | −0.34 | −0.36 | −0.08 | 0.05 | 0.34 * | 0.68 * | 0.68 * | 1.00 |  |  |  |  |
| POPUL | 0.21 * | 0.22 * | −0.12 * | −0.21 * | −0.53 | 0.45 * | −0.23 | −0.70 * | 1.00 |  |  |  |
| LEXP | −0.80 | −0.81 * | 0.30 * | 0.45 | 0.26 * | 0.21 | 0.18 | 0.21 * | 0.06 | 1.00 |  |  |
| REG | −0.18 | −0.14 | −0.40 | −0.49 * | −0.41 * | −0.60 | −0.43 * | −0.40 | 0.44 | 0.00 * | 1.00 |  |
| STAB | −0.34 * | −0.33 | −0.64 | −0.52 * | −0.22 | −0.41 * | −0.15 * | 0.15 * | −0.12 | 0.32 | 0.57 * | 1.00 |

**Source:** Authors. * Statistically significant at 5%.

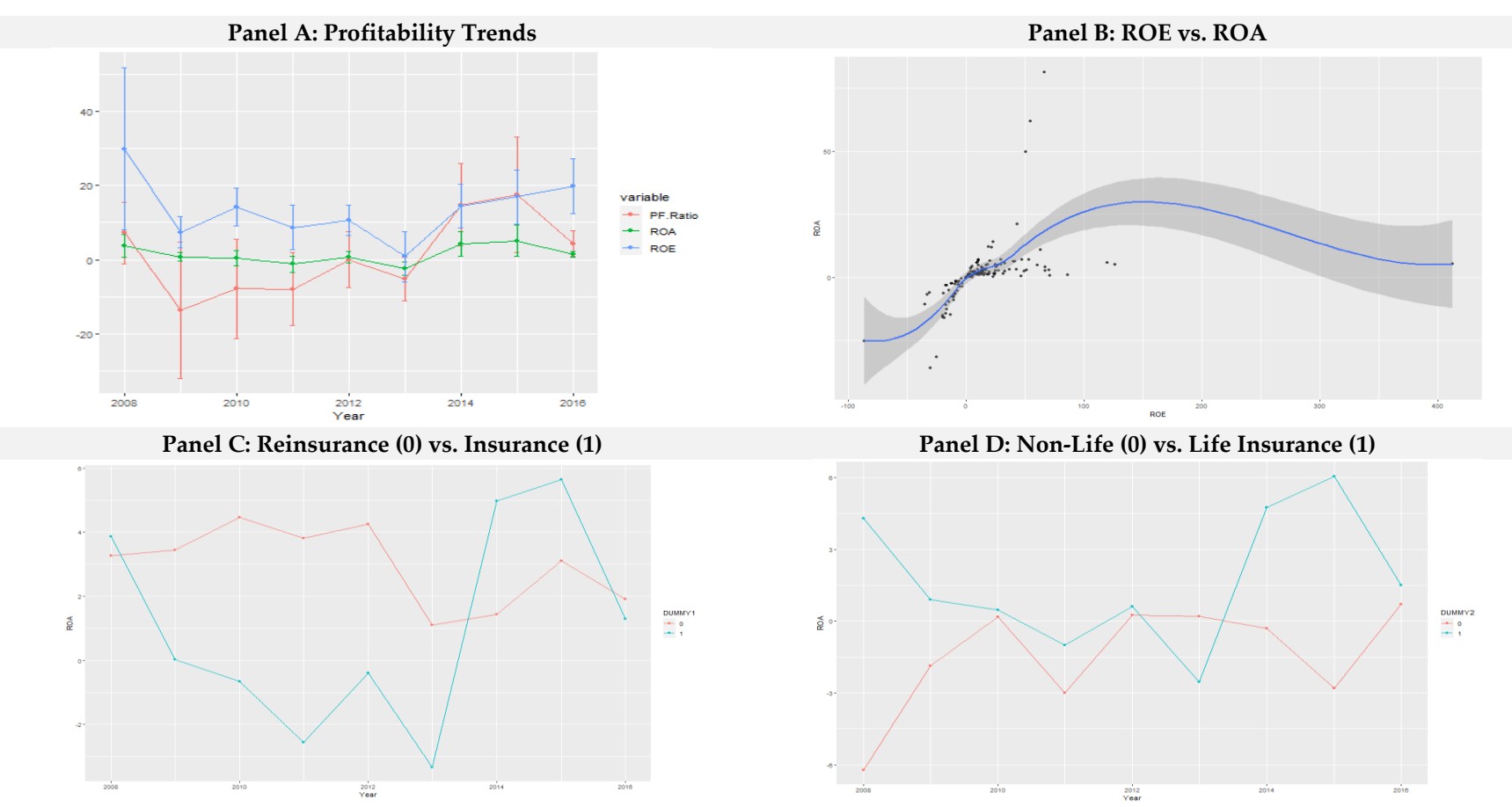

**Figure A1.** Profitability dynamics. Source: Authors.

Table A3. Stationarity testing.

| Level | | | | | | | | | | | | | | | | | |
|---|---|---|---|---|---|---|---|---|---|---|---|---|---|---|---|---|---|
| | ROA | | ROE | | ROTP | | SIZE | | RISK | | HHI | | GDP | | INFL | | WAGE |
| TEST | Stat. | P* | Stat. | P* | Stat. | P* | Stat. | P* | Stat. | P* | Stat. | P* | Stat. | P* | Stat. | P* | Stat. | P* |
| Test 1 | −10.9 | 0.00 | −1.7 | 0.04 | −5.0 | 0.00 | 1.8 | 0.96 | −1.38 | 0.08 | −3.13 | 0.00 | 6.52 | 1.00 | 0.76 | 0.78 | −2.06 | 0.02 |
| Test 2 | −3.5 | 0.00 | −1.7 | 0.04 | −2.5 | 0.01 | 3.0 | 1.00 | 0.24 | 0.60 | 0.58 | 0.72 | 5.03 | 1.00 | 3.36 | 1.00 | −0.13 | 0.45 |
| Test 3 | 77.8 | 0.00 | 64.5 | 0.00 | 67.8 | 0.00 | 6.4 | 1.00 | 25.04 | 0.91 | 21.45 | 0.97 | 2.00 | 1.00 | 5.42 | 1.00 | 29.32 | 0.78 |
| Test 4 | 98.3 | 0.00 | 108.0 | 0.00 | 100.0 | 0.00 | 7.6 | 1.00 | 20.14 | 0.98 | 83.76 | 0.00 | 0.69 | 1.00 | 20.76 | 0.98 | 81.65 | 0.00 |

| Level (Continued) | | | | | | | | | | |
|---|---|---|---|---|---|---|---|---|---|---|
| | POPUL | | LIFEEXP | | REG | | STABILITY | | Explanation | |
| TEST | Stat. | P* | Stat. | P* | Stat. | P* | Stat. | P* | Variable | Description | Variable | Description |
| Test 1 | −2.32 | 0.01 | −8.45 | 0.00 | −7.28 | 0.00 | −6.81 | 0.00 | ROA, ROE, ROTP | Stationary | POPUL | Non-stationary |
| Test 2 | 3.23 | 1.00 | −0.32 | 0.38 | −1.15 | 0.12 | 0.19 | 0.58 | SIZE, RISK, HHI | Non-stationary | LIFEEXP | Non-stationary |
| Test 3 | 5.82 | 1.00 | 31.54 | 0.68 | 42.64 | 0.21 | 25.59 | 0.90 | GDP, INFL | Non-stationary | REG | Non-stationary |
| Test 4 | 1.34 | 1.00 | 12.58 | 1.00 | 142.6 | 0.00 | 17.24 | 1.00 | WAGE, SAVING | Non-stationary | STABILITY | Non-stationary |

| First-Difference | | | | | | | | | | | | | | | | | |
|---|---|---|---|---|---|---|---|---|---|---|---|---|---|---|---|---|---|
| | SIZE | | RISK | | HHI | | GDP | | INFL | | WAGE | | POPUL | | LIFEEXP | | REG |
| TEST | Stat. | P* | Stat. | P* | Stat. | P* | Stat. | P* | Stat. | P* | Stat. | P* | Stat. | P* | Stat. | P* | Stat. | P* |
| Test 1 | −6.00 | 0.00 | −5.91 | 0.00 | −24.0 | 0.00 | −10.6 | 0.00 | −2.59 | 0.00 | −12.5 | 0.00 | −13.3 | 0.00 | −2.86 | 0.00 | −36.6 | 0.00 |
| Test 2 | −2.75 | 0.00 | −0.14 | 0.44 | −8.10 | 0.00 | −6.38 | 0.00 | −3.10 | 0.00 | −4.10 | 0.00 | −3.74 | 0.00 | 0.51 | 0.69 | −13.3 | 0.00 |
| Test 3 | 69.40 | 0.00 | −6.67 | 0.00 | 143.8 | 0.00 | 120.8 | 0.00 | 48.90 | 0.07 | 88.51 | 0.00 | 83.02 | 0.00 | −2.95 | 0.00 | 203.5 | 0.00 |
| Test 4 | 93.53 | 0.00 | 58.95 | 0.01 | 120.1 | 0.00 | 221.7 | 0.00 | 127.4 | 0.00 | 128.1 | 0.00 | 148.7 | 0.00 | 127.3 | 0.00 | 13.83 | 1.00 |

| First-Difference (Continued) | | | | | | | | |
|---|---|---|---|---|---|---|---|---|
| | STABILITY | | Explanation | | | | | |
| TEST | Stat. | P* | Variable | Description | Variable | Description | Variable | Description | Explanation |
| Test 1 | −7.81 | 0.00 | SIZE | Stationary | INFL | Stationary | LIFEEXP | Stationary | All the variables are stationary at first-difference |
| Test 2 | −4.93 | 0.00 | RISK | Stationary | WAGE | Stationary | REG | Stationary | |
| Test 3 | 100.5 | 0.00 | HHI | Stationary | SAVING | Stationary | STABILITY | Stationary | |
| Test 4 | 1.84 | 1.00 | GDP | Stationary | POPUL | Stationary | ——— | —— | |

**Source:** Authors. * Probabilities for Fisher tests are computed using an asymptotic chi-square distribution. All other tests assume asymptotic normality. Test 1: Levin, Lin, and Chu test. Test 2: Im, Pesaran, and Shin W-stat. Test 3: ADF—Fisher chi-square. Test 4: PP—Fisher chi-square.

**Table A4.** Consistency/efficiency tests (LM, F-test, Hausman test).

| Model 1: ROA | | | |
|---|---|---|---|
| Lagrange Multiplier Test (Random Effects vs. POLS) | | | |
| Normal | df. | *p*-value | Winner |
| 2.7952 | ————————- | 0.002594 | Random Effects |
| Alternative hypothesis: significant effects | | | |
| F test for individual effects (Fixed Effects vs. POLS) | | | |
| F-value | df1/df2 | *p*-value | Winner |
| 1.0389 | 17/119 | 0.4223 | POLS |
| Alternative hypothesis: significant effects | | | |
| Hausman test (Fixed effects vs. Random effects) | | | |
| Chi-squared | df | *p*-value | Winner |
| 9.5317 | 7 | 0.2167 | Random Effects |
| Alternative hypothesis: one model is inconsistent | | | |
| Model 2: ROTP | | | |
| Lagrange Multiplier Test (Random effects vs. POLS) | | | |
| Normal | df. | *p*-value | Winner |
| 5.5155 | ————————- | $1.739 \times 10^{-8}$ | Random Effects |
| Alternative hypothesis: significant effects | | | |
| F test for individual effects (Fixed effects vs. POLS) | | | |
| F-value | df1/df2 | *p*-value | Winner |
| −0.55819 | 17/119 | 0.9999 | POLS |
| Alternative hypothesis: significant effects | | | |
| Hausman test (Fixed effects vs. Random effects) | | | |
| Chi-squared | df | *p*-value | Winner |
| 10.8214 | 7 | 0.1466 | Random Effects |
| Alternative hypothesis: one model is inconsistent | | | |

**Source:** Authors.

**Table A5.** Mixed-effects model (Model 4 : $\beta_{BS} \neq \beta_{WS}$).

| | Panel A: ROA | | | Panel B: ROTP | |
|---|---|---|---|---|---|
| Variable | Coefficient | Prob. | Variable | Coefficient | Prob. |
| TIME | 0.003 | 0.000 *** | TIME | 0.004 | 0.000 *** |
| $SIZE_{BS}$ | 0.744 | 0.003 ** | $SIZE_{BS}$ | 0.852 | 0.012 ** |
| $SIZE_{WS}$ | −0.015 | 0.010 ** | $SIZE_{WS}$ | −0.183 | 0.005 *** |
| $RISK_{BS}$ | −0.212 | 0.015 ** | $RISK_{BS}$ | −0.354 | 0.013 *** |
| $RISK_{WS}$ | 0.006 | 0.025 ** | $RISK_{WS}$ | 0.131 | 0.041 ** |
| $DUMMY_{BS}$ | —- | —- | $DUMMY_{BS}$ | —- | —- |
| $DUMMY_{WS}$ | 1.231 | 0.006 *** | $DUMMY_{WS}$ | 1.012 | 0.000 *** |
| $HHI_{BS}$ | −0.155 | 0.001 ** | $HHI_{BS}$ | −0.422 | 0.014 ** |
| $HHI_{WS}$ | 0.088 | 0.044 ** | $HHI_{WS}$ | 0.104 | 0.038 ** |
| $GDP_{BS}$ | 0.312 | 0.022 ** | $GDP_{BS}$ | 0.139 | 0.002*** |
| $GDP_{WS}$ | 0.123 | 0.001 *** | $GDP_{WS}$ | 0.157 | 0.000 *** |
| $INFL_{BS}$ | −0.201 | 0.024 ** | $INFL_{BS}$ | −0.137 | 0.029 ** |
| $INFL_{WS}$ | 0.114 | 0.000 * | $INFL_{WS}$ | 0.044 | 0.036 ** |
| $WAGE_{BS}$ | 1.166 | 0.329 | $WAGE_{BS}$ | 0.946 | 0.221 |
| $WAGE_{WS}$ | −0.068 | 0.185 | $WAGE_{WS}$ | 0.458 | 0.197 |
| $POPUL_{BS}$ | 0.559 | 0.041 *** | $POPUL_{BS}$ | 0.321 | 0.000 *** |
| $POPUL_{WS}$ | −0.148 | 0.002 ** | $POPUL_{WS}$ | −0.133 | 0.033 ** |
| $LEXP_{BS}$ | −0.203 | 0.299 | $LEXP_{BS}$ | −0.428 | 0.233 |
| $LEXP_{WS}$ | −0.137 | 0.111 | $LEXP_{WS}$ | 0.137 | 0.176 |
| $REG_{BS}$ | −0.744 | 0.477 | $REG_{BS}$ | −0.422 | 0.109 |
| $REG_{WS}$ | 0.228 | 0.256 | $REG_{WS}$ | −0.218 | 0.424 |
| $STAB_{BS}$ | 0.592 | 0.007 ** | $STAB_{BS}$ | 0.455 | 0.003 *** |
| $STAB_{WS}$ | −0.255 | 0.000 * | $STAB_{WS}$ | −0.185 | 0.002 *** |
| R-squared | 0.211 | | R-squared | 0.244 | |
| Adj. R-sq | 0.203 | | Adj. R-sq | 0.236 | |
| S.E. Reg. | 1.277 | | S.E. Reg. | 1.133 | |
| B-P LM test * | 11.6621 (*p* = 0.2267) | | | 9.1844 (*p* = 0.1791) | |
| Pesaran CD * | 1.2021 (*p* = 0.3869) | | | 1.5685 (*p* = 0.4452) | |
| B-G/W test * | 17.8435 (*p* = 0.2755) | | | 19.2701 (*p* = 0.2849) | |
| B-P heter. * | 1.8642 (*p* = 0.2691) | | | 1.6481 (*p* = 0.3193) | |

* Breusch–Pagan LM and Pesaran CD test of independence, Breusch–Godfrey/Wooldridge test for serial correlation, and the Breusch–Pagan test for heteroskedasticity. Source: Authors. Note: Significance at the 10%, 5%, and 1% level are denoted by ***, **, *, respectively.

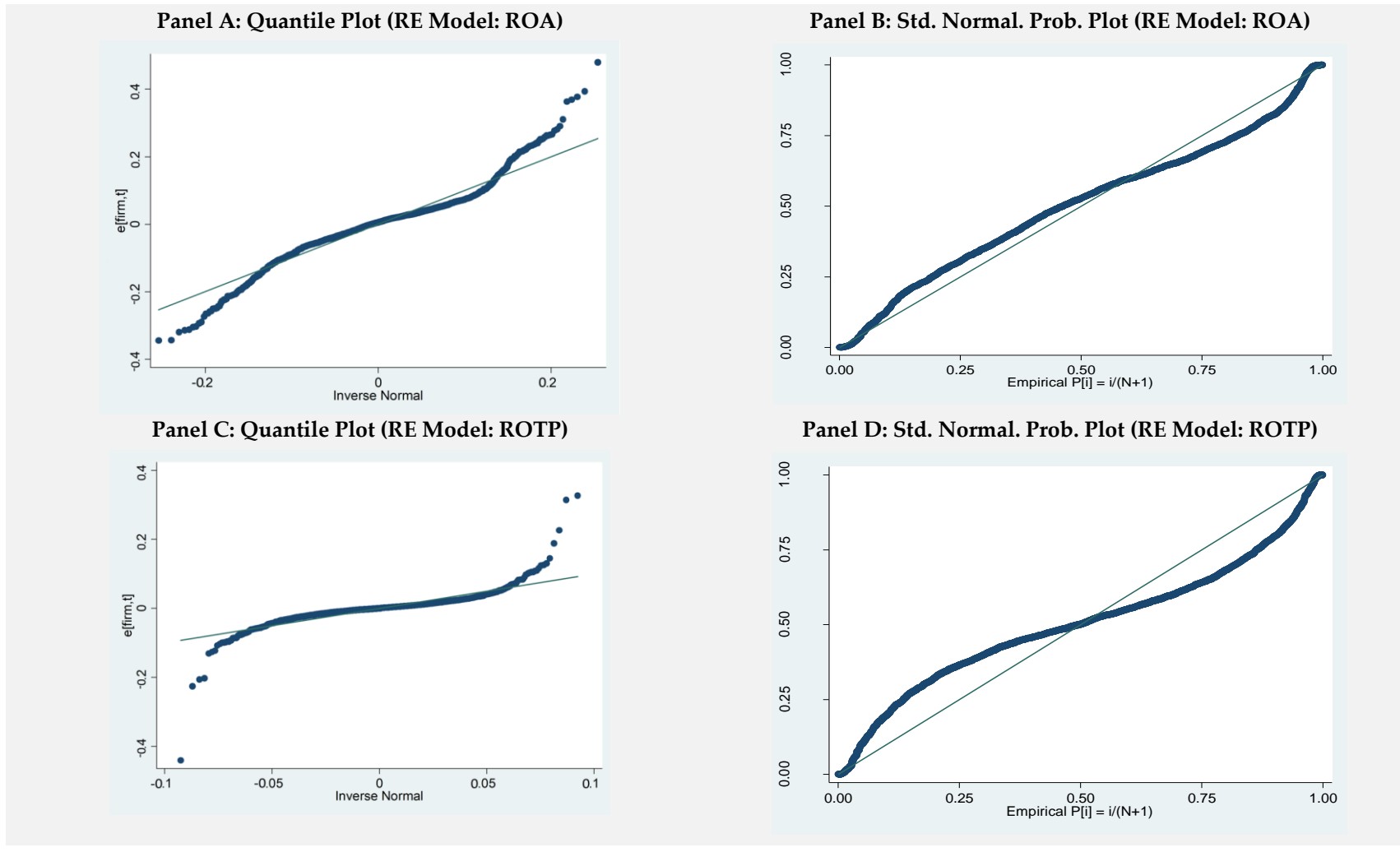

**Figure A2.** Model diagnostics. Source: Authors.

**Table A6.** GMM estimation.

| | Dependent Variable: ROA | | | | | Dependent Variable: ROTP | | | |
| | Panel A: FD GMM | | Panel B: System GMM | | | Panel C: FD GMM | | Panel D: System GMM | |
| Variable | Coefficient | Std. Err. | Coefficient | Std. Err. | Variable | Coefficient | Std. Err. | Coefficient | Std. Err. |
|---|---|---|---|---|---|---|---|---|---|
| c | 1.3354 | 1.1501 | 0.7632 | 0.5311 | c | 1.2133 | 0.9431 | 0.8644 | 0.6466 |
| $\Delta$ROA(lagged) | 0.0506 | 0.0082 *** | 0.0829 | 0.0060 *** | $\Delta$ROTP(lagged) | 0.1284 | 0.0269 *** | 0.1012 | 0.0483 ** |
| SIZE | 0.1130 | 0.0184 *** | 0.1554 | 0.0460 *** | SIZE | 0.0985 | 0.0461 ** | 0.1843 | 0.0736 ** |
| RISK | −0.1125 | 0.0480 ** | −0.1847 | 0.0563 *** | RISK | −0.1065 | 0.0342 *** | −0.1598 | 0.0513 ** |
| DUMMY | 0.0312 | 0.63612 | 0.0519 | 0.1842 | DUMMY | 0.0166 | 0.0614 | 0.0222 | 0.0503 |
| HHI | −0.0901 | 0.1202 | −0.0231 | 0.0555 | HHI | −0.0808 | 0.1263 | −0.0122 | 0.0189 |
| GDP | 0.09338 | 0.0072 *** | 0.05881 | 0.00131 *** | GDP | 0.0838 | 0.0152 *** | 0.0509 | 0.0211 *** |
| INFL | 0.0391 | 0.0131 | 0.0601 | 0.4747 | INFL | 0.0238 | 0.0141 * | 0.0509 | 0.0365 |
| WAGE | 0.0211 | 0.01361 | 0.0757 | 0.0494 | WAGE | 0.0476 | 0.0729 | 0.0159 | 0.0136 |
| POPUL | 0.2759 | 0.0605 *** | 0.1566 | 0.0237 *** | POPUL | 0.1621 | 0.0366 *** | 0.0856 | 0.0297 |
| LEXP | 0.2963 | 0.5758 | 0.0587 | 0.13759 | LEXP | 0.2451 | 0.2041 | 0.0638 | 0.0437 |
| REG | 0.0231 | 0.0125 * | 0.0305 | 0.1642 | REG | 0.0977 | 0.1310 | 0.1183 | 0.1498 |
| STAB | 0.1108 | 0.0478 ** | 0.0658 | 0.0051 *** | STAB | 0.1456 | 0.0251 *** | 0.0712 | 0.0311 ** |
| Adj. R-sq | 0.1203 | | 0.1444 | | Adj. R-sq | 0.0821 | | 0.1033 | |
| S.E. Reg. | 0.1928 | | 0.2451 | | S.E. Reg. | 0.1331 | | 0.1567 | |
| J-stat (Prob.) | 15.2153 (0.1942) | | 29.4624 (0.1152) | | J-stat (Prob.) | 13.2153 (0.2166) | | 27.4481 (0.0922) | |
| A-B (1) z-st (Prob.) | −5.3351 (0.0000) | | −3.3956 (0.0000) | | A-B (1) z-st (Prob.) | −4.6822 (0.0000) | | −3.1145 (0.0000) | |
| A-B (2) z-st (Prob.) | −0.3428 (0.7344) | | −0.4867 (0.5211) | | A-B (2) z-st (Prob.) | −0.4921 (0.5466) | | −0.6288 (0.3787) | |

**Source:** Authors. Note: Significance at the 10%, 5%, and 1% level are denoted by ***, **, *, respectively.

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
