# Peer review of "Determinants of Sustainable Profitability of the Serbian Insurance Industry: Panel Data Investigation"

_sustainability, doi:10.3390/su14095190_

Round 1

Reviewer 1 Report

Dear authors, Your paper devoted to the investigation of sustainable profitability in the Serbian market is an interesting and statistically well-established paper. However, there are some sections that need to be improved. 

  1. Could you please unify, if the analysis was realized in the period 2008-2016 (as mentioned in the abstract) or 2008-2019 (as indicated in the text of the paper)? Why was this period chosen? please add in the paper the information why this period is so important for the analysis.
  1. Introduction - it this section the background of the topic is briefly described, as well as the main aim and the hypotheses. If possible, please, highlight the novelty of your study.
  1. LR - please add more European-based studies published recently which are focused on the financial performance determinants, e.g. Slovak and Czech environments (10.24136/eq.2021.023 or 10.24136/oc.2020.006), etc.
  1. When 3 or more authors are cited, not all their names are displayed - only the first one + et al.
  1. formula (1) - it is a generally known algorithm which does not need to be presented. 
  1. the results of the paper are very interesting and supported by relevant international studies. That is why I would recommend renaming this section to "results and discussion" and adding more studies to enable you to compare / confront/ discuss your findings with other relevant studies. 
  1. improve the quality of Fig. 1 - it is not possible to read it.
  1. the discussion section, as it is in the current version of the paper, is very weak - it must be improved, as indicated in the text above. 

9. conclusions should be rewritten, please, focus of your crucial findings, theoretical contributions, practical implication, limitations of your research and future research challenges. 

10. there are so many grammar mistakes, the paper must be proofread. 

11. table A2 - it is not clear, which correlation is statistically significant and which not - please add the information. 

Reviewer 2 Report

The article is logical and consistent. Conclusions result from the conducted research. The research methods were selected correctly. 

It is not quite understandable, why the results of the studies for the period 2008-2016, are not published until now. Five years have passed, which in the case of changing markets is of great importance. This should be clarified in the introduction. The research from the given period are a bit outdated. 

In addition, no source is provided in the tables.

The date of access to internet sources has not been given. 

Reviewer 3 Report

The significance/robustness of the findings/conclusions could be enhanced if the following details of the sample studied were more explicitly stated:

  1. What is % of the Serbian insurance market covered by the 18 companies studied?
  2. How many of the companies studied are in life insurance?

A brief description of the Serbian Insurance Market evοlution over the period studied could be useful as well

Reviewer 4 Report

General:

The first comment concerns causation. In this paper, the Authors very often use the term “impact” when the Authors only uses the regression type methodology. The correct solution is to use methods based on the concept of impact effects. The use of econometric methods is a good substitute choice due to the inability to perform counterfactual analyzes. It is best to use the terms "relationship" or "association".

The comment concerns the statistical significance of variables in regression models, to which the Authors attach great importance. In empirical research, it is often the case that statistical significance is not that important. I believe that this is not defective. It is part of the current debate of scientists from various fields (excessive belief in p-value).

  1. Abstract:Is abstract structured in accordance with the requirements of the Journal?

The topic of the paper is very interesting, however the paper has some limitations.  

The abstract is a bit confusing, and it is not in accordance with the title of the paper. The abstract provides a summary of the paper. The Authors could better highlight the contribution of the paper in the abstract.

The abstract should contain the paper’s objective, motivation, methodology, main findings, and contribution.

  1. Motivation and contribution:Does the paper contain new and significant information that improve or build on existing research? What is theoretical, empirical and/or practical contribution of the paper?

The introduction is clear, however, it is worth highlighting (more clearly!) the research hypothesizes/questions and I did not find the main results obtained in the empirical analysis. I think that it is important to anticipate the main results in this part of the paper in order to allow the reader to better understand the analysis.

  1. Literature: Does the paper present an adequate understanding of the relevant literature and refer to literature sources in the right way?

The lack of previous actual studies (in 2019, 2020, 2021, 2022). Please add and capture previous studies about this topic You need also highlight the finding from previous studies and try to tell the different approach of your study and then illustrate the contribution and novelty from your study

  1. Methodology: Is the paper's idea built on an appropriate conceptual background? Is the research design and methods employed appropriate?

The research methodology is straightforward.

The sample structure is not shown. Equation (1) is incorrect missing individual effects.

In the explanation for  only index t is used. In equations (2), (3)  appears, and where ?

Additionally, there is the problem of endogeneity of variables, the above problem is not solved by the basic models used in the work. There is VAR in the keyword, no VAR at work.

Additionally, it is worth taking into account the dynamics of the behavior of variables.

  1. Findings: Are the results presented in a clear way, highlighting the novelty in the context of existing research?

Model diagnostics not performed.

  1. Quality of Communication: Is the paper written in a clear way for readers and the used terminology is appropriate?

It is clearly written.

Round 2

Reviewer 1 Report

Dear authors, thank you for making the significant changes, which improved the overall quality of the paper. 

Author Response

Thank you for your valuable suggestions and comments.

Reviewer 2 Report

I accept the corrected article. 

Author Response

(The authors gave the same response as above.)

Reviewer 4 Report

Dear Authors,

Thank you very much for the corrections made to the article.

In the current version there is already a more precisely focused literature review, but we are writing about the risk, about the pandemic, and there are still too few literature on this subject. For ex,.

COVID-19: stress-testing non-financial companies: a macroprudential perspective. The experience of Poland, Eurasian Economic Review 11 (2), 283-319, 2021

Climate Risk with Particular Emphasis on the Relationship with Credit-Risk Assessment: What We Learn from Poland, Energies 14 (23), 8070, 2021
